# Unreasonable effectiveness of LLM reasoning: a doubly cautionary tale of temporal question-answering

**Dagmara Panas**                                          *daga.panas@ed.ac.uk*
*School of Informatics*
*University of Edinburgh, UK*

**Ali Payani**                                              *apayani@cisco.com*
*Cisco Research*

**Vaishak Belle**                                          *vbelle@ed.ac.uk*
*School of Informatics*
*University of Edinburgh, UK*

**Reviewed on OpenReview:** *https://openreview.net/forum?id=1DkDONd8Rd*

## Abstract

The remarkable success of Large Language Models in modeling both the syntax and the semantics of language has prompted a body of research into language-adjacent abilities, most notably commonsense reasoning. As LLMs' performance continues to advance on successive benchmarks, we turn to temporal reasoning, which lags somewhat behind other tasks due to its more complex logic. We start from previous work, where authors successfully induce (apparent) reasoning by breaking down the problem into a two-step procedure of temporal graph extraction and subsequent reasoning. Specifically, in the first step an LLM is prompted to parse a natural language description into a semi-structured timeline of events; and in the second step, it is given the extracted timeline and prompted to answer a temporal reasoning question. We conjecture that this procedure presents two separate opportunities for introducing errors and further hypothesise that a Neuro-symbolic approach should help in this matter. We follow the recent trend of using external executors in concert with LLMs to carry out exact reasoning and verification. We see the reasoning step of the original two-step procedure as a natural target for a symbolic solver and design a rule-based solution for Temporal Question-Answering, drawing on ideas from Allen's Interval Algebra. To our surprise, we find that our rule-based reasoner does *not* improve beyond the previously reported, purely neural solution. It appears that both our approach and the previous method operate at around the limits of achievable performance, imposed by the correctness of information extraction. Such a result seems to suggest that a non-symbolic LLM is capable of symbolic-level reasoning, although upon further investigation we discover that not to be the case. It is not that the neural solution makes no reasoning mistakes, but rather that the LLM manages to compensate for some of its erroneous replies by *short-cutting* to the correct answer in other questions; a.k.a. not reasoning but guessing. Although the effect is not pronounced performance-wise, we feel it is conceptually important: as we argue, production of correct answers is *not* a measure of reasoning.

## 1 Introduction

The unprecedented success of Large Language Models in modeling natural language now includes simulating convincingly human-like chain-of-thought verbalisations of reasoning (Wei et al., 2022b), tool use (Schick et al., 2024), and planning (Huang et al., 2022). Importantly, although trained in a primarily unsupervised

paradigm, these architectures are capable of few-shot generalisation to new tasks (Brown et al., 2020), raising questions of emergent capabilities (Wei et al., 2022a). Further, in the short time since their introduction, LLMs continue to be refined: billions of parameters turn to tens of billions, additional strategies such as mixture-of-experts add to their strengths (Jiang et al., 2024), and data added to the training pool continues to fill the gaps at the edges of the distribution. Hand in hand with such improvements comes widespread interest in assessing LLMs, from conversational Turing-tests (Turing, 1950; Sejnowski, 2023) to a range of language comprehension and reasoning benchmarks (Srivastava et al., 2022; Mittal et al., 2024; Duan et al., 2024; Parmar et al., 2024). Research is now coming out at speeds that defy the ability of individual researchers to keep up with the field. Thus, ironically to the subject matter, it would seem that an AI system capable of natural language comprehension and reasoning might be what is required to understand the current state of LLMs. Indeed, efforts in that direction have already begun, with researchers proposing fully-automated scientific discovery approaches (Lu et al., 2024). It may be tempting to resort to such tools, particularly as the newest generation of models is marketed as capable of 'reasoning' (Chollet, 2024). However, while undeniably impressive in action, these models are underwritten by a purely neural paradigm, and as such are inherently limited. This fact is well understood among the proponents of Neuro-Symbolic integration (Besold et al., 2021), but also increasingly appreciated more widely (Schaeffer et al., 2024; Pfister & Jud, 2025; Stechly et al., 2025; Lee et al., 2024; Valmeekam et al., 2022; Shojaee et al., 2025). The key issue with the data-driven mode of automated decision-making is that outputs are an opaque blend of data, inductive priors, and chance. As a result, no rule is fixed, and there are no hard guarantees on getting the correct answers, or even ones that are self-consistent (Bao et al., 2024). Perhaps the most salient demonstration of that fickle nature comes from the recent PromptReport, which details an investigation into the 'black art' of prompt engineering (Schulhoff et al., 2024). Authors surface some unusual observations such as that an accidental duplication of part of the prompt can result in improved performance; or that automatic prompt optimisation yields as the most effective a 'word salad' jumble of tokens rather than coherent language. Unsurprisingly then, we and others (Giunchiglia et al., 2025) argue that systems without guarantees are inadmissible in safety-critical applications. Neuro-symbolic integration is not, however, straightforward to achieve. One common avenue to leverage rigorous rule-based systems is tool-augmented LLMs, with 'flavours' such as retrieval-augmented generation (RAG) or python-assisted LLMs (PAL), as well as more specialised ones such as LogicLM (Hu et al., 2022) or ToM-LM (Tang & Belle, 2024). The premise of this approach to integration is that the neural model invokes an external, verifiable tool to perform some or all of the reasoning. In essence, it amounts to acting as a parser between natural language and tool-specific syntax such as Python, Prover9 (McCune, 2005–2010), etc; potentially also relaying back the answers in natural language. For areas with ample training data, such as Python, this can work remarkably well for a majority of common / simpler problems. However, for highly specialised areas such as sound logical reasoning, success tends to be limited, particularly as problem complexity increases (Pan et al., 2023). Nevertheless, the tool-invoking / LLM-as-a-parser route provides an attractive alternative to relying on approaches such as chain-of-thought (CoT)(Wei et al., 2022b) and its variants, which *rationalise their guesses* rather than *reasoning to their answers* (Bao et al., 2024; Stechly et al., 2025). We are interested in this line of *hybrid NeSy* work, and specifically in applying it to temporal reasoning problems.

Although temporal logic is common in everyday life and abundant in natural language, LLMs still struggle with this task (Chen et al., 2021; Fatemi et al., 2024; Tan et al., 2023). The difficulty stems from the fact that reasoning about time requires the application of both logic and arithmetic, and also often commonsense / world knowledge. As Xiong et al. (2024) and others (Su et al., 2024; Tan et al., 2023) demonstrate, it is feasible to improve temporal reasoning within a purely neural paradigm. However, there remains room for improvement, both in terms of performance, but also conceptually and computationally. We aim to demonstrate the value of symbolic and rule-based approaches to address this gap by extending the study of Xiong et al. (2024). We identify the work as a promising starting point because the authors propose to break down the problem into two separate steps: temporal information extraction; and reasoning over the extracted *temporal graphs*. Such a formulation is a natural candidate for hybrid NeSy, as we expect LLMs to be better at extracting information than at reasoning; and, unlike natural language, temporal graphs are suitable inputs for a rule-based reasoner. We implement a custom solution to Temporal Question Answering in Python, inspired by Allen's Interval Algebra (AIA) (Allen, 1983), which attains a near-perfect score on the ground truth temporal graphs. We then evaluate against the original (Xiong et al., 2024) on the same

benchmarks (TimeQA (Chen et al., 2021), TempReason (Tan et al., 2023), TGQA (Xiong et al., 2024)), while retaining the LLM in the function of a parser. Surprisingly, we find no improvement over the reported bootstrapped CoT answers, yet a marked improvement over 'vanilla' CoT. Considering our results on the true temporal graphs, we suspect a form of *short-cutting* is taking place in the LLM. By short-cutting we refer here to the phenomenon of language models selectively attending / responding to the prompt tokens, which can result in ignoring explicit constraints, and further in paradoxical provision of correct answers even when fed false premises. To verify our concerns we employ a similar approach to those of Bao et al. (2024) and Stechly et al. (2025), who investigate a related shortcoming of LLMs, namely models providing a reasoning chain that is inconsistent with the final answer. Bao et al. (2024) compare correctness of the CoT against the model's ultimate response, having first either automatically or manually scored the chain-of-thought reasoning steps. To their surprise, over 60% of simple arithmetic problems suffer from short-cutting-like issues – answering correctly despite erroneous intermediate calculations. Stechly et al. (2025) also check CoT's consistency with the final response, but they go further than that: in addition, they also train a transformer architecture with task-irrelevant as well as erroneous chain-of-thought samples. Again, counter-intuitive to expectations, they find that performance on the task improves vs. training without CoT, even though the additional information in the training prompts was *not useful* by design. Thus, in a similar vein, we further examine the extracted temporal graphs as well as the individual answers of the LLM-based reasoner, with our symbolic AIA-based solution as the verifier of temporal graphs. Specifically, we flag instances where the responses are correct despite the temporal graph itself being faulty. We discover these cases to constitute approximately 12% of the dataset, confirming our suspicions. Our results thus add to the growing body of research challenging the deceptive nomenclature and marketing of LLMs as capable of reasoning or thinking (Bao et al., 2024; Schaeffer et al., 2024; Stechly et al., 2025). Particularly problematic in this context is the predominant focus on correct performance, which, as we show, can be artificially inflated. Arguably, Neuro-symbolic integration is harder than purely connectionist architectures in terms of engineering effort, as well as being inflexible to slight task variations. Further, it does not necessarily offer performance improvements, as in our case. We pose here that such difficulties should not be taken as a cautionary tale against NeSy; on the contrary, they confirm it to be at least a valuable verification tool, if not a target of aspiration of AI as a field.

## 2 Related work

A number of recent works address the problem of temporal reasoning in language models, noting the particular challenge posed by the logic of time; we focus here on the most directly related ones (Chen et al., 2021; Fatemi et al., 2024; Su et al., 2024; Tan et al., 2023; Xiong et al., 2024). First, Chen et al. (2021) introduce TimeQA, one of the earliest benchmarks designed specifically for *temporal*, or *time-sensitive* question answering (TQA), and one which we include in our evaluation. Since the study predates the latest advancements in LLMs, their solution is to train a custom question-answering encoder-decoder transformer architecture. Although as a purely task-dedicated model it is of less relevance to present work on 'generalist' LLMs, it provides useful context for our results; they achieve between 0.47–0.6 accuracy.

Second, Tan et al. (2023) propose TempReason, a more comprehensive benchmark, explicitly aimed at advancing temporal reasoning capabilities of LLMs. Of particular relevance is the fact that they introduce a variant to the typically assumed open-book question answering, whereby rather than providing information to the model in natural language, they provide only the structured facts. Their solution involves pre-training, fine-tuning, and further time-sensitive reinforcement learning, the last one requiring a reference structured timeline of events. This significantly improves over baselines, particularly in the mode of evaluation with structured data in the prompt. These improved scores are between 0.15–0.21 accuracy, which underscores the difficulty of temporal reasoning. Further, it should be noted that for each type of temporal reasoning identified by the authors (time-time, time-event, and event-event relationship questions) a separate fine-tuning is applied.

The following study of Xiong et al. (2024), takes the idea of leveraging a structured timeline (Tan et al., 2023) further. The authors introduce TG-LLM, a two-step pipeline for TQA that formally splits the task of reasoning over natural language into, first, temporal information extraction; and second, reasoning. Since datasets featuring natural language and corresponding structured timelines are rare, Xiong et al. (2024) also develop a synthetic dataset, TGQA, which also features new temporal questions. To test the transferrability

of their approach and synthetic data, they also use TempReason (Tan et al., 2023) and TimeQA (Chen et al., 2021) datasets. They propose Low-Rank Adapter (LoRA) (Hu et al., 2022) fine-tuning for the medium-sized open-source LLM Llama (Touvron et al., 2023), separately for the first and second steps of the workflow. Specifically, the first adapter requires training with ground truth temporal graphs as *target*, in order to extract a structured timeline of events from free text in the proposed TG-LLM solution. The second adapter is trained with ground truth temporal graphs as *input*, and answers to temporal queries as the target for prediction. Once both are trained, they are meant to be used in a consecutive fashion, with first adapter's output feeding into the second as context alongside the question (see Figure 1 in Methods). We note that since each step requires true graphs, (Xiong et al., 2024) need to train a pair of adapters in each setting. That is, a separate pair for each of TGQA, TempReason and TimeQA benchmark datasets; but also a separate pair for the TGQA-to-TempReason transfer and the TGQA-to-TimeQA transfer[1]. In other words, their transferrability investigation applies primarily to the first adapter. An additional aspect of TG-LLM involves modifications of the fine-tuning process for the 'reasoning' adapter: chain-of-thought bootstrapping, external knowledge injection, and graph augmentation. The first method involves generating multiple high-quality CoTs for each sample, to sample from for inclusion with the standard supervised fine-tuning prompt. The second modification assists 'reasoning' by including helpful information in the prompt (presumably derived from the temporal graph) such as '1999 < 2000'. The final addition addresses the issue of data scarcity, and related lack of variety, which may harm performance when the trained adapter is to be used on inferred rather than ground truth graphs. Thus they propose to increase the sample size and diversity by augmentations such as changing all dates by an offset or replacing entity names. Against the 'vanilla' CoT (Wei et al., 2022b) baseline, which does not involve temporal graph extraction merely a single fine-tuning, they achieve an accuracy improvement of between 0.07–0.3. Compared to the basic TG-LLM, the 'full' bootstrapped + knowledge-enhanced + augmented TG-LLM improves by approx. 0.25. With these improvements the performance on TGQA is around 0.8, similarly to GPT-4. As outlined already, since we expect LLMs to be better information extractors than reasoners, we also expect that the majority of the remaining gap is due to reasoning errors. Thus we conjecture that swapping the second adapter for a rule-based reasoner should further improve the overall results.

We take note also of two other recent benchmarks, even though we do not utilise them in our work. First of these, the Temporal Reasoning for Large Language Models (TRAM) (Wang & Zhao, 2023) dataset is a comprehensive assessment of temporal understanding and reasoning. The benchmark consists of over 520k samples and spans 38 tasks, which includes a variety of arithmetic, reasoning, factual and narrative / causal questions. Although extensive in the range of topics, it is limited in that the TRAM queries are cast as multiple-choice questions. The study provides also an evaluation of a number of models, noting the variability in performance across tasks and the remaining gap to human performance.

The second notable benchmark is the Test-of-Time (ToT) proposed by Fatemi et al. (2024). Unlike the majority of previous efforts, which tend to draw on existing databases and resources such as WikiData (Vrandečić & Krötzsch, 2014), ToT is an entirely synthetic dataset. Particularly interesting is its *semantic* portion, aimed at reasoning over timelines of events. Random graphs of various structures and sizes are generated programmatically and their nodes and edges are used to fill a text template of the form 'E1 was the R1 of E2 from 1999 to 2003.'. Entities and relations remain fully anonymous, distinguished by numericals. This truly isolates the task of reasoning about time from any linguistic correlations that may exist in the training data of LLMs. It is of particular interest to us, as such data structure supports a fully rule-based reasoner; we return to this in the Discussion.

## 3 Datasets

Since the aim is to extend previous work, we use the same datasets as the original TG-LLM paper (Xiong et al., 2024): TimeQA (Chen et al., 2021); TempReason (Tan et al., 2023); and the newly introduced TGQA (Xiong et al., 2024). These are all temporal question-answering datasets, intended for evaluating and / or fine-tuning LLMs, and thus each sample naturally consists of a minimum of: a short piece of narrative text; a

---

[1]That is presumably due to the fact that TGQA questions differ substantially from TempReason and TimeQA, and the second of the adapters needs to be fine-tuned to the benchmark being evaluated.

Table 1: Summary of the datasets used. Notation: 'TGn', number of unique temporal graphs; 'TGl', average length of temporal graphs; 'Qn' number of questions (of a given type); 'Q type', type of queries in the benchmark; 'RQn' number of questions removed. Note, removal of ambiguous-answer questions was prior to calculating the other statistics.

| Dataset | TGn | TGl | Qn | Q type | RQn |
|---|---|---|---|---|---|
| TimeQA, easy | 746 | 4.0 | 2913 | 'What happened from *Time1* to *Time2*?' | 73 |
| TimeQA, hard | 746 | 4.0 | 2461 | 'Which event was happening at *Time1*' | 104 |
| | | | 502 | 'Which event happened before/after *Time1*?' | |
| TempReason, l2 | 1052 | 5.8 | 5153 | 'What happened in *Feb YYYY*?' | 28 |
| TempReason, l3 | 1002 | 5.6 | 4126 | 'Which event happened before/after *EventX*?' | 72 |
| TGQA | 99 | 5.8 | 753 | 'Given the following K events ... which was the *j*th?' | 409 |
| | | | 332 | 'Which event happened before/after *EventX*?' | |
| | | | 303 | 'Which started first, *EventX* or *EventY*?' | |
| | | | 303 | 'Were *EventX* and *EventY* in the same year?' | |
| | | | 212 | 'Did *EventX* overlap *EventY*?' | |
| | | | 188 | 'Was *EventX* longer than *EventY*' | |
| | | | 292 | 'How much time between *EventX* and *EventY*?' | |
| | | | 221 | 'How long did *EventX* last? | |
| | | | 303 | 'When did *EventX* occur?' | |

question requiring the understanding of time / temporal logic; and the correct answer. However, since Xiong et al. (2024) propose to leverage the concept of *temporal graphs* to guide reasoning, they (and therefore we) also require access to the correct, structured timeline of events. Note, we use *temporal graphs* as a shorthand for the structured timeline of events following Xiong et al. (2024), although none of the datasets adhere to the typically assumed (subject, relation, object, start, end) syntax. Both the TempReason and the TGQA dataset contain the ground truth temporal graphs, owing to the construction of these benchmarks (see below). For TimeQA, Xiong et al. (2024) separately extract the temporal graphs, employing a mix of rule-based and LLM-based processing. Following a semi-automatic verification step confirming their robustness, these are then treated as the ground truth[2]. We evaluate performances on the *test* split of each. We summarise in Table 1 and describe each dataset below. Illustrative examples are included in the Appendix.

TimeQA (Chen et al., 2021) draws on WikiData (Vrandečić & Krötzsch, 2014) for accurate time-sensitive facts[3] and Wikipedia for their corresponding text. It is constructed in a semi-automated manner that combines data mining and crowd-sourced verification. Each sample text can have between 2–8 questions associated with it. As per original classification, questions concern four temporal relations: 'in'; 'between'; 'before'; and 'after'. Importantly, the dataset supports two levels of difficulty, *easy* vs. *hard*, distinguished by the level of explicitness. Specifically, the question might pertain to a date or date range explicitly mentioned

---

[2]For convenience we download all datasets from a repository provided by Xiong et al. (2024), rather than from their respective original sources.
[3]That is, facts that involve a date or a temporal period, e.g. 'John Doe was born in 1880'.

in the text, or it might require an answer that is implied by other dates, without being included *verbatim* in the story. Thus effectively the *easy* portion of TimeQA is more a test of temporal understanding than temporal reasoning. It may be useful to point out here that the original classification into 'relations' by Chen et al. (2021) does not map cleanly to those of Allen's Interval Algebra. In the context of our work, we find it more helpful to classify instead as in Table 1. All the *easy* questions need merely to identify which event is associated with a given date (range). The *hard* queries need in addition the carrying out of temporal logic, and its formal relations, such as 'before' or 'during'. In our development we find that, being semi-automatically generated, this benchmark occasionally contains issues; e.g. name 'Unknown' is used both as valid entity with explicitly stated dates (e.g. in Table 3), and as a fall-back answer for time outside of the temporal graph.

TempReason (Tan et al., 2023) also leverages the knowledge base of WikiData (Vrandečić & Krötzsch, 2014), and similarly selects time-sensitive facts for generating the benchmark. However, as it is a more contemporary study aware of previous efforts, its approach departs from TimeQA and results in a somewhat different set of questions. Although there is some overlap in terms of the main subject of temporally-evolving facts, there is less than 1% overlap in terms of (subject, answer) pairs, indicating that the benchmarks are complementary. Importantly, the creation includes a convenient step of selecting sets of facts connected by the subject, which means the temporal graph is already provided with the data. In TempReason a single timeline can have between 1–38 questions associated with it, and it tends to be on average 2 nodes longer than TimeQA's (see Table 1). This benchmark supports 3 variants of temporal reasoning questions[4], named *l1*, *l2* and *l3*. In contrast to TimeQA, these subtypes are not explicitly linked to difficulty, but rather to the type of task. Namely, the variants differ by asking about, respectively: a time-time relation; a time-event relation; and, an event-event relation. Following Xiong et al. (2024) we only use the *l2* and *l3* subtypes. Also this benchmark contains the occasional problematic sample, e.g. overlapping nodes.

Finally, TGQA (Xiong et al., 2024) is a synthetic dataset created by sub-sampling the Yago11k knowledge graph and then generating corresponding narratives in natural language, using GPT-3.5. Random sub-graphs of up to 5 nodes are selected, and, to avoid potential issues of data memorisation in pre-trained LLMs, anonymised. To ensure alignment between the produced story and the source temporal graph, an additional semi-automated validation step is introduced. An LLM is queried for each individual temporal fact; should it fail to produce the correct response, the sample is then inspected manually. Although the source data is a temporal knowledge graph, Xiong et al. (2024) choose to provide the structured timeline in a manner that separates the start and end date of events, to aid downstream LLM-based reasoning. In addition, unlike TimeQA and TempReason which focus on a single subject and multiple objects of a relation, TGQA admits symmetric nodes. For example, 'A married B' and 'B married A' both get included in the timeline. This is why the average length of temporal graphs is over 5 (see Table 1), despite sampling only up to 5 nodes from Yago11k. Questions are then generated by populating templates from the ground-truth temporal graph. This graph-first rather than story-first construction of TGQA allows for a slightly wider variety of question types than either TimeQA or TempReason. Specifically, Xiong et al. (2024) distinguish 9 categories of queries, requiring different forms / combination of temporal relations and arithmetic, e.g. ordering, deciding overlap / simultaneity, duration etc. The full list is provided in Table 1. With such construction, each temporal graph has between 16–70 questions associated with it. While beneficial for having varied queries, we note that this means only 99 temporal graphs are used as input. Finally, also this benchmark suffers from problematic samples, only the issue is not exactly minor. The symmetric nodes are not treated as equivalent in situations where they theoretically should, being simultaneous and thus in identical temporal relation to other nodes. This issue also affects other nodes that are not semantically symmetric, but still are temporally equivalent (e.g. two events with different subjects and objects happening at exactly same time). In these instances, even though two answers are valid, TGQA provides only one as the ground truth, seemingly arbitrarily (we suspect it may prioritise the 'main character' of the story, but do not investigate further). Since this affects 12% of the dataset, we choose to remove from each dataset the samples where one of 2 simultaneous events is given as the ground truth; we note these numbers in Table 1.

Overall, we note that all of these are relatively simple as benchmarks of temporal reasoning. In majority of the cases the timeline concerns a single individual's history, and the average length is only between 4–5.8

---

[4]TempReason also contains questions that do not test reasoning, but e.g. knowledge of temporal facts.

nodes. Questions for 2 of the 3 benchmarks are fairly limited in variety, and while TGQA probes for more diverse relations and even some arithmetic, a sizable portion of the questions only require a yes/no answer. However, as performance of LLMs in even such straightforward scenarios is lacking, we see these as suitable for our purposes.

## 4 Methods

As illustrated in Figure 1, the original study leveraged a large language model both for extracting the TG from text, as well as reasoning over it. We propose to compare a symbolic reasoner to the LLM, when provided *inferred* temporal graphs.

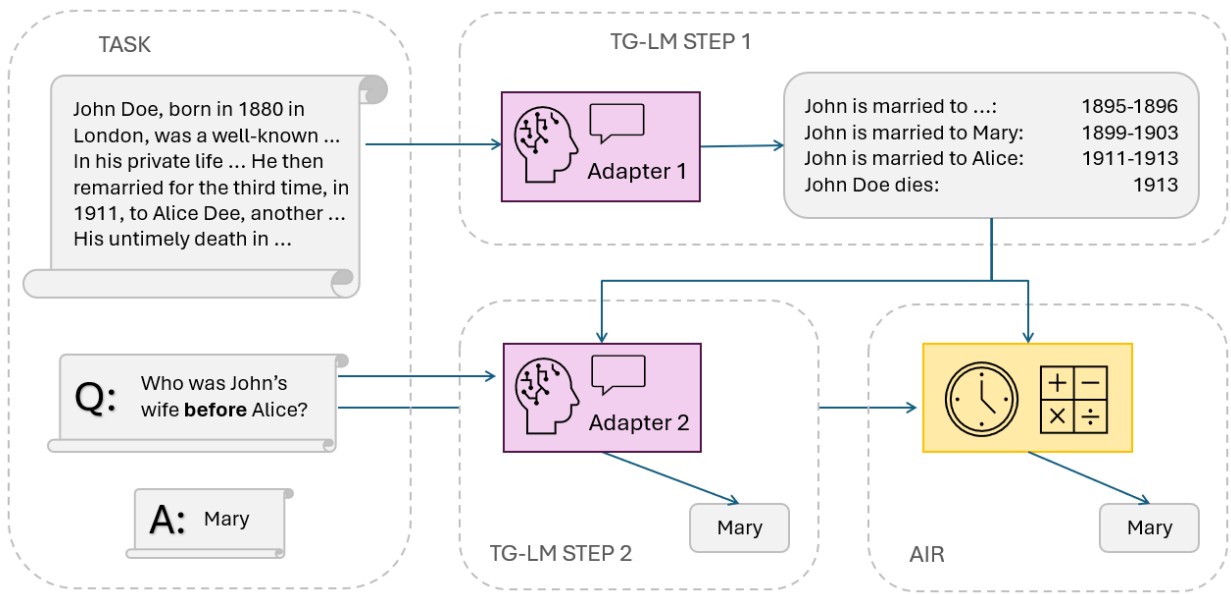

Figure 1: Left, the task formulation, a triplet of: narrative description of events; question requiring the understanding of time; and the correct answer. Top right, the first step of the 2-step pipeline of (Xiong et al., 2024): a fine-tuned LLM extracts the temporal graph from natural language. Bottom middle, the second step of the 2-step pipeline: a separate fine-tuned LLM answers questions based on structured text, rather than full narrative. Bottom right, our proposed alternative second step: a Python rule-based reasoner answers questions based on structured text and the rules of Allen's Interval Algebra.

### 4.1 Rule-based Temporal Question-Answering

We develop a minimal rule-based question-answering code base in Python, leveraging the relations from Allen's Interval Algebra (AIA)(Allen, 1983). AIA formalises the commonsense temporal logic around the notion of intervals and their possible relations. Allen (1983) posits 13 such relations (see Table 2 for selected examples, or Table 4 in Appendix A.2 for all 13), which together with the transitivity table supports development of algorithms for reasoning about events in time. We do not implement the full language of Allen's algebra simply because we do not require it for our current purposes, though we see it as a fruitful direction for future work.

The core element in our processing pipeline is a simple Python *Interval* class [5] that implements AIA relations as operations that can be performed between the two instances of the class. Specifically, each instance requires start and end times to be provided on initialisation; this allows each of the AIA relations to be an invokable method returning Boolean truth values as to whether the relationship holds. For example, to understand

---

[5]Implementation adapted from the publicly available https://github.com/bartonip/pyintervals/tree/master

Table 2: Selected relations of Allen's Interval Algebra, to illustrate the premise.

| Relation | Notation | Inverse | Illustration | Implementation |
|---|---|---|---|---|
| X before Y | $<$ | $>$ | XXX
YYY | X.end < Y.start |
| X meets Y | $m$ | $mi$ | XXX
YYY | X.end == Y.start |
| X overlaps Y | $o$ | $oi$ | XXX
YYY | X.start < Y.start & X.end > Y.start |

whether event X has occurred before event Y, the dedicated method checks whether the end time of X was before the start time of Y (first row of Table 2).

To perform verifiable reasoning, the supporting code handles parsing of the plain-text temporal graphs into Python objects, as well as question-answering. First, each TG is encoded as an unordered collection of instances of the *Interval* class. Simple string parsing extracts the start and end times from the structured text, as well as the names / descriptions of the events as labels. The parsing utilities are entirely tailored to the 3 benchmarks, both in terms of date formats, and node naming. To align with dataset assumptions and similarly to Xiong et al. (2024), where available we leverage the knowledge of entity names. Second, since different questions of temporal logic will require the evaluation of different AIA relations (or their combinations), separate sub-routines handle answering each possible question type. Heuristic rules then match questions to appropriate function streams. For an illustrative example, see Figure 2: since the keyword 'before' occurs in the text of the query, the subroutine that checks for either 'before' or 'meets' will be selected. All intervals will be then compared to the target 'Alice' one, and those evaluating to *True* selected. From these, the one chronologically last will be selected. We note that we make a number of simplifying assumptions in developing our code base; e.g. in this example we ignore the possibility that events may overlap. We refer to our rule-based TQA solution as Allen's Interval Reasoner (AIR).

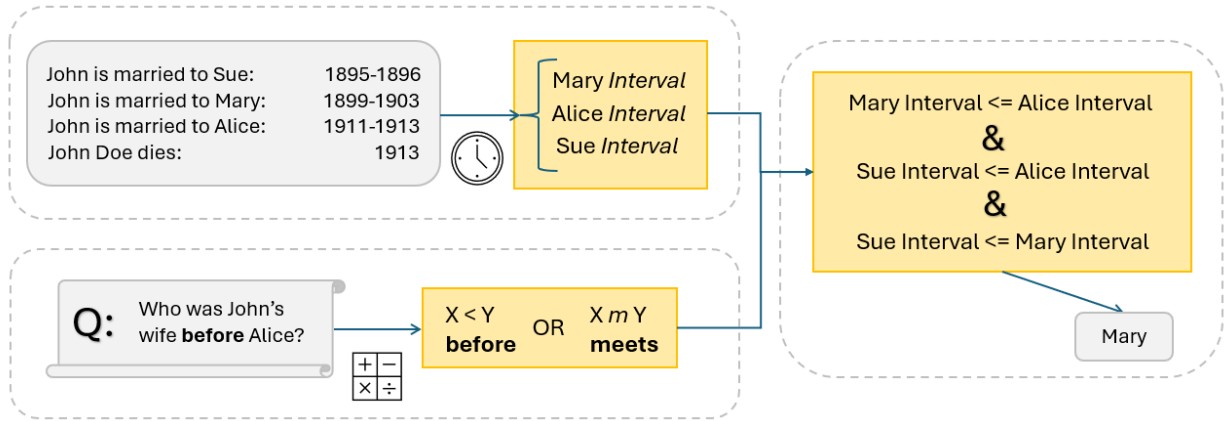

Figure 2: Top left: parsing the structured text to obtain an unordered collection of objects of the *Interval* class. Bottom left: matching question types to appropriate AIA relation-based question-answering strategy. Right: query is answered by processing the collection of intervals with the selected utility.

We note that the code base does not provide a universal temporal reasoner since neither of the datasets provide instances of true knowledge graphs, but rather forms of structured text. Thus, we merely develop a

set of utilities that cover the question types and the temporal graph formats present in the three benchmarks (TimeQA, TempReason, TGQA); many more are possible. Furthermore, the set of utilities is developed on the *true* temporal graphs (TG-true), a setting where parsing and QA are straightforward due to their well-structured nature. However, with the aim being to apply the AIR system on *inferred* graphs (TG-pred), we need to consider the fact that LLM-extracted TGs may contain formatting errors. Although, following Xiong et al. (2024) we will be using an instruction to 'Return json only', it is well known that LLMs may over-generate, repeat tokens, etc. We therefore also develop utilities to correct all such mistakes. Admittedly, this is not a general or easily scalable solution for a NeSy system; the process requires substantial time, and only works for the model and task context of the study. The present study was meant as a proof-of-concept of an AIA-LLM system, although it (inadvertently) turned into a cautionary tale of deceptive effectiveness of LLMs when evaluated on performance only. We leave a more general extension to future work, as outlined in the Discussion.

## 4.2 Inferred temporal graphs

Since the datasets provided by Xiong et al. (2024) do not contain the *inferred* temporal graphs, only the ground truth ones, we need to reproduce a portion of the study. Specifically, we need to train appropriate low rank adapters (LoRA) (Hu et al., 2022) for temporal graph extraction, and then run these in inference mode on the held-out *test* data. Recall from Section 2 that in their original approach, Xiong et al. (2024) need to train a separate first (a.k.a. text-to-TG) adapter for each setting: the 3 benchmarks; and the 2 TGQA-to-TempReason and TGQA-to-TimeQA transfer. Since we are both resource-constrained, and primarily interested in a real-world setting, where curated training data is not available, we choose to train on the smaller, synthetically-generated TGQA. We use the code provided (Xiong et al., 2024) and similarly use the open-source Llama-2 13B model (Touvron et al., 2023) due to limited resources. We still have 3 separate adapters for each of the datasets, to cater to their varying TG formats. Each benchmark specifies the graph in a different manner (see Table 3 in A.1), and the format of the TG that is being output by the first adapter needs to match the format of the input to the reasoning step (be it LLM or AIR). Thus, we use the same *train* split TGQA input stories and true temporal graphs for each of the 3 supervised fine tuning runs, only vary how the target TG is formatted. Finally, to produce inferred temporal graphs, we apply the respective fine-tuned adapter to *test* split of TimeQA, TGQA and TempReason. We follow Xiong et al. (2024) in all settings to the best of our knowledge. That means we include an in-context example at test time and use the 'easy' inference mode.

## 4.3 Evaluation

To evaluate AIR-LLM responses we use exact match (EM), as our rule-based reasoner returns nodes of the TG rather than free text. However, since in our string parsing we employ the knowledge of relevant entities, and we additionally 'clean up' LLM answers, we compare to perplexity-based accuracy of Xiong et al. (2024). Perplexity-based accuracy selects from relevant entities the one with lowest perplexity vs. LLM-generated response. This allows for ignoring any formatting issues or rephrasing by LLM, and thus makes for a fairer comparison to ours. We evaluate our rule-based reasoner on the ground truth temporal graphs and reach performance between 0.95 and >0.99. We note that for the 2 datasets where accuracy is not, as may be expected, near-perfect, from manual inspection we ascertain that many of the errors are due to particularities of the benchmarks. As mentioned already, e.g. TimeQA admits ambiguous use of the entity 'Unknown'. We do not examine all of AIR's failure cases, as for demonstrative purposes of the work we deem 0.95 sufficient.

We also develop a simple scoring function to evaluate the inferred temporal graphs against the ground-truth ones (TG-pred-score). This serves as a sanity check for the evaluation of results on the *predicted* temporal graphs. Our question answering AIR system respects temporal logic by design, and the results from ground truth temporal graphs assure us to the correctness of parsing and routing utilities. However, with the LLM outperforming our symbolic solver, we want to additionally verify that AIR is correct to the degree the graphs themselves are. Further, since we suspect short-cutting, we will use the scoring function as an aid in testing our hypothesis. The scoring function grants 0.5 point for each correct start and end of a temporal interval in the graph, and averages the total over the length of the TG. For example, if the timeline consists of 4 events and the LLM extracts the timing of only 2 correctly, the score will be 0.5; it will also be 0.5 if all events'

start times are correct, but end times incorrect. We also have a variant that penalises over-generation, since additional nodes in a temporal graph may impact on the correctness of answers. For brevity we omit that from discussion, as it does not impact materially on the results.

### 4.4 Additional analyses and experiments

Observing surprisingly poor outcomes against the reported LLM-based reasoning, and wanting to better understand the reasoning discrepancies, we investigate further. First, to complement the TG-pred-score function, we also devise a heuristic check for what we term *malformed* predictions. The graph scoring aggregates together all issues of the LLM-based timeline extraction. That means wrong but coherent timelines, but also cases where returned text can not be parsed into intervals at all. To assure ourselves this is not merely our oversight in post-processing, and to distinguish the two types of issues, we therefore check for several conditions indicative of unparseability; for example, complete lack of numerals, or using a suspiciously large or small set of characters, when compared to the true TGs.

Second, we also decide to re-run the second step of the TG-LLM pipeline (Xiong et al., 2024). We again fine-tune a LoRA adapter (Hu et al., 2022) for Llama-2 13B (Touvron et al., 2023) on the train split of TGQA, with the ground truth temporal graphs as input, and reasoning answers as output. We do not use CoT bootstrapping nor graph augmentation for computational reasons, but otherwise reproduce Xiong et al. (2024); that is, we use only the 'base' / 'vanilla' Chain-of-Thought (Wei et al., 2022b) and provide additional information in the prompt. With adapter trained, we run it both on the inferred, as well as ground truth temporal graphs. Of particular interest to us is the run on *predicted* TGs, as this gives us access to the individual answers of the LLM, and allows for verification if short-cutting is taking place. We only carry out this experiment on TGQA, for computational and researcher time considerations.

## 5   Results

### 5.1   A surprising AIR drop

We develop the rule-based utilities for temporal question answering, AIR, around the notion of Allen's Interval Algebra. As mentioned already, we limit ourselves to the question types and TG formats present across TimeQA, TempReason and TGQA datasets. We achieve $\geq 0.95$ accuracy when applying AIR on the ground truth TGs, as shown in the top row of Figure 3. However, when applied to the *inferred* temporal graphs, our reasoner drops sharply in performance. The only dataset where scores remain above 0.5 is TGQA; the one used for fine-tuning the story-to-graph adapters. On the remaining TempReason and TimeQA benchmarks the performance is very poor, ranging between 0.17 and 0.3, as can be seen in the second row of Figure 3. Importantly, and to our surprise, these scores are all between approx. 0.1 and as much as 0.37 **below** the reported, purely neural solutions (we re-report these as TG-LLM (full)* and GPT-4 in Figure 3, from the original Xiong et al. (2024)). When compared to the best scores from the original benchmark papers (Chen et al., 2021; Tan et al., 2023), only on the *l2* split of the more difficult TempReason AIR-LLM manages to narrowly outperform the baseline. Seeing as we achieve excellent / near-perfect performance on true TGs, we therefore suspect that the issue lies with the *inferred* temporal graphs.

### 5.2   Reasoning quality vs. information quality

To isolate TG quality from reasoning quality, we also report our scoring of temporal graphs against the ground truth. This is the third row in Fig. 3, denoted as TG-pred-score. Reassuringly, these align with the accuracies of AIR-LLM's answers, with at most 0.06 difference in favor of TG-pred-score. Thus, either the inferred graphs for TempReason and TimeQA are severely lacking at between 0.19–0.32, or our parsing of LLM's answers is. Conscious that the inferred temporal graphs are at risk of formatting issues, we employ additional utilities to correct such minor mistakes. Of course, it is difficult to assure complete coverage without manual inspection of each sample; however, we discover in the process of spot-checking the predicted TGs with a score of 0 that many are malformed beyond repair. These are, conveniently for us, rather glaring departures from coherent text, as illustrated in Figure 4 A. Therefore we are able to employ another set of simple heuristics to quantify the proportion of timelines in TempReason and TimeQA that

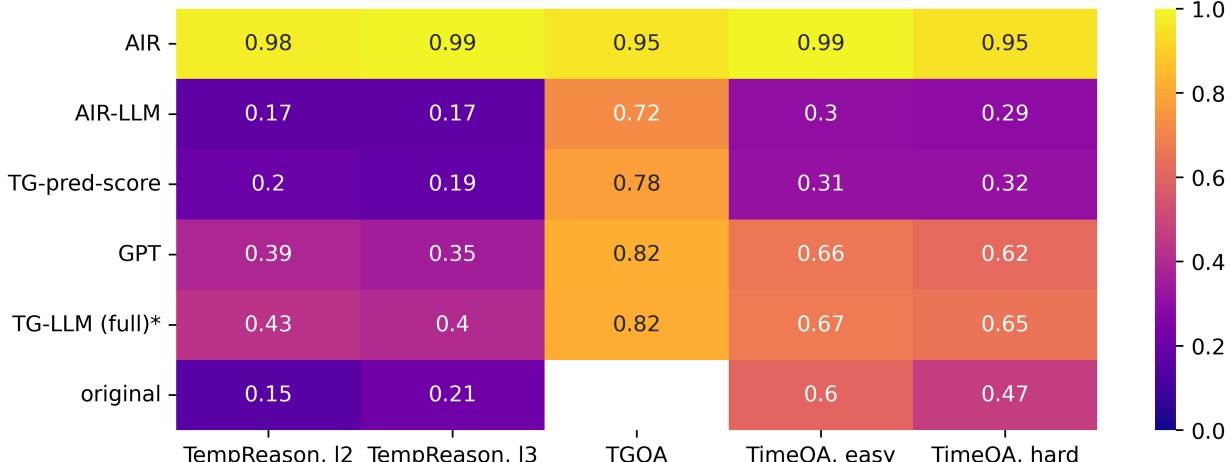

Figure 3: Temporal question answering and graph quality results. Rows (all EM accuracy, unless noted otherwise): 'AIR' and 'AIR-LLM', accuracy of our solution on TG-true and TG-pred respectively; 'TG-pred-score' quantifies the quality of extracted TGs, relevant for understanding AIR-LLM accuracy; 'GPT-4', 'TG-LLM (full)*' are re-reported from Xiong et al. (2024), with '*' to denote that for TG-LLM we report perplexity-based accuracy, as explained in the Methods; 'original' from Tan et al. (2023) and Chen et al. (2021) for TempReason and TimeQA, respectively. Columns: datasets, with split where applicable.

are merely a meaningless collection of characters / tokens, or otherwise significantly broken. These account for practically all of the empty TGs (see Fig. 4 B), with which we conclude our parsing is not at fault.

Thus, we are able to answer practically all questions when provided true graphs, and we also know the inferred graphs to be lacking. Further, looking at TG-pred-score against the performance of TG-LLM, particularly for TimeQA and TempReason, we see a puzzling pattern. Despite the information content of predicted TGs being between 0.19–0.32, TG-LLM is nevertheless able to answer correctly at the level of 0.4–0.67 accuracy. Admittedly, it is still possible to answer a number of TQA queries correctly, even if most of a given graph is incorrect, provided the question pertains to the correct portion of the TG or the error does not impact on the temporal function queried. For example, if all the intervals in a graph are offset by a set amount of time, neither their chronological order nor duration would be affected. However, we doubt such convenient happenstances can account for all observed gains; not to mention TG-LLM would need to simultaneously not make any reasoning errors on the correctly extracted TGs. We are therefore led to conclude that the performance gap between our AIR-LLM and the reference TG-LLM of Xiong et al. (2024) may at least in part (if not in entirety) be due to short-cutting.

### 5.3 Unreasonably good LLM reasoning

To verify our suspicions, we require further analyses. The first piece of evidence we happen upon when counting the proportion of malformed predictions. We note from Figure 4 B that as many as 36% of samples in TimeQA suffer this issue. This means that the limit of achievable performance, whatever the question, is strictly 64%, as there is *no usable information* in the predicted timeline of the rest of the data. When we return to Fig. 3 we find that TG-LLM yet manages to score 0.67. That is a discrepancy of only 0.03, but that minority 0.03 of questions would only be answerable through pure guesswork, a.k.a. short-cutting. However, given that we are comparing our re-run of TG extraction against scores reported in Xiong et al. (2024), it is still technically possible that the few percent difference might be accounted for by chance. Different runs of LLMs are not guaranteed to yield the same results, and the proportion of malformed predictions may have been lower in the original study. Thus for a definitive answer regarding short-cutting we need the individual responses of an LLM reasoner when fed *our* extracted temporal graphs.

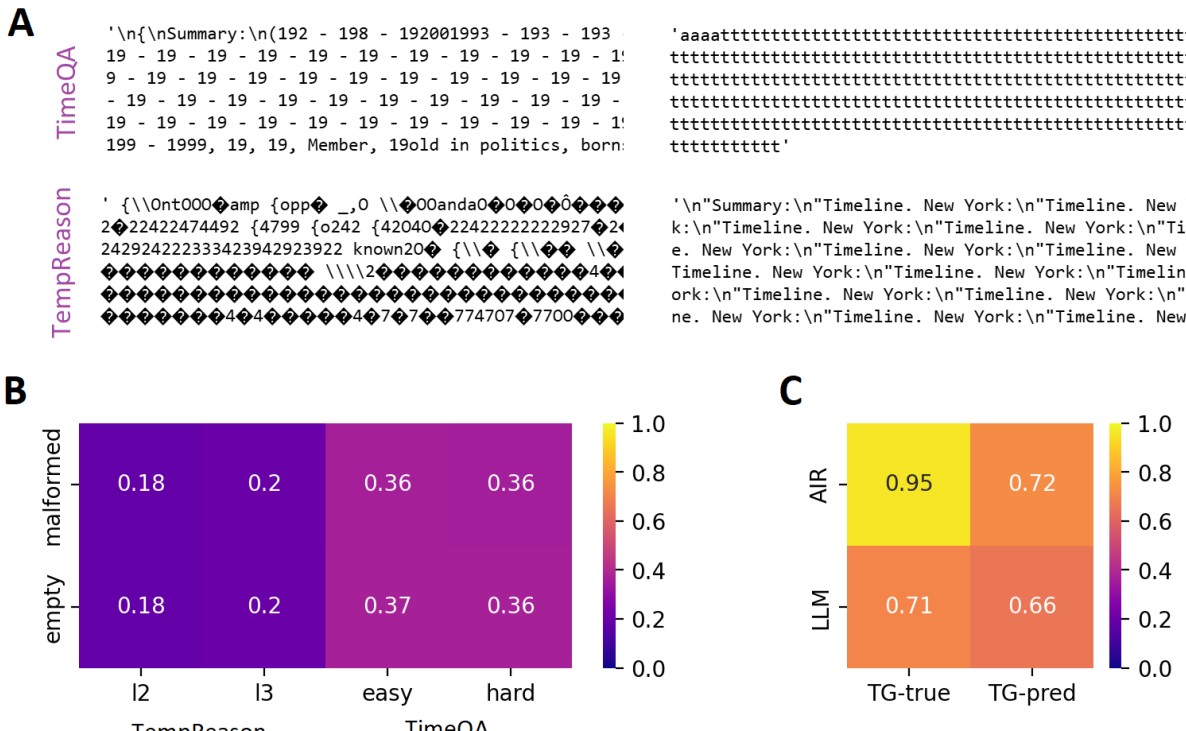

Figure 4: A: Two examples of mal-formed predictions, from TimeQA and TempReason benchmarks; since the adapter was trained on TGQA we expect the most glaring mistakes to be made on the other benchmarks. B: Proportion of inferred temporal graphs that are empty compared to the proportion of text-to-TG LLM responses that are malformed. C: Comparison of the rule-based vs. LLM-based reasoning on true vs. predicted temporal graphs.

We select the TGQA dataset for the in-depth case study and re-run the second portion of the TG-LLM pipeline (Xiong et al., 2024), in its' simplest form. That is, we fine-tune a *reasoning* adapter on true temporal graphs (from the *train* split), and then apply it to the *test* split TGs *inferred* by the first adapter. For completeness, we also run the reasoning adapter on true graphs from the *test* split, to isolate TG-LLMs reasoning performance from graph extraction performance. Before delving into individual answers, we first look at the overall performance of TG-LLM against AIR-LLM. Two things are apparent from the results shown in Figure 4 C. First, without CoT bootstrapping or TG augmentation of the main results of Xiong et al. (2024), the performance of the neural reasoner no longer surpasses our symbolic AIA-based solution, dropping from 0.82 to 0.66. Second, even when provided the **ground truth** temporal graphs, LLM's performance is lower than the reported TG-LLM on **inferred** TGs by entire 11% (compare the 0.71 from Fig. 4 C, cell TG-true LLM, with the 0.82 in Fig. 3, cell TGQA TG-LLM*). Though still not conclusive proof, both of these observations are in line with our short-cutting hypothesis. The fact that removing data / fine-tuning augmentation methods degrades performance is no surprise: LLMs are not *reasoning* towards the answer, but rather *predicting*, hence their responses are liable to change with the data or manner of posing the question (Schulhoff et al., 2024). The second observation underscores the point of LLMs merely providing 'educated guesses': in the 'TG-true' setting the input data *has all the information one needs to reason to the answer*; the only way one could get any of these wrong (beyond the odd ambiguity of the question), much less 30%, is if one were guessing. Even if there were no short-cutting to be detected, we still find these results problematic: obviously LLMs are not reasoners, but give them the right hints, drill them the right way, and they can seem like ones.

Finally, we investigate whether the LLM is just unreasonably good, or impossibly so, a.k.a. if the 0.66 performance on our inferred TGs is *genuine.* Specifically, we check whether any of the correct LLM answers coincide with *incorrect* AIR-LLM answers. We proceed to find that for approx. 12% of the dataset the LLM short-cuts: it answers correctly, despite the inferred temporal graph being faulty and implying a wrong answer. We further check that for all of these cases, the predicted timeline does not contain one of the events needed to provide the answer. Generally, predicted timelines are shorter than the true ones, which often affects only one end of an interval. Since 12% corresponds to over 350 question-answer pairs, we do not perform a manual verification of all of these, only spot-checks. From these we want to highlight an interesting case study example, which we encounter several times. It involves a situation where the question asks about a node that does not exist in the timeline verbatim. However, although the event is not in TG-pred, the timeline does contain something very similar; e.g. TG-true states 'John Doe was born in York in 1888' (and this is also the subject of the TQA question), while TG-pred contains 'John Doe was born in *Stirling* in 1888'. TG-LLM's answer to the question is 1888, which *is* the ground truth answer; and thus by its' associative nature TG-LLM's second adapter manages to undo the extraction mistake that the first adapter made. Though we posit this should not count as correct *reasoning* performance, and still constitutes short-cutting, we believe it is not necessarily a behavior to be eradicated. We return to this matter in the concluding section.

## 6 Discussion

We set out to demonstrate the benefit of employing a hybrid NeSy approach for the task of temporal reasoning in natural language, an area where the complementary strengths of neural and symbolic approaches are particularly salient. Since the logic of time is somewhat more complex than e.g. commonsense reasoning, and specialised data less abundant, it is an area still ripe for improvements for LLMs. At the same time, temporal reasoning in a verifiable manner is relatively straightforward with appropriately structured input. We develop such a rule-based reasoner for time-sensitive question-answering as a natural complement of the TG-LLM framework (Xiong et al., 2024). Specifically, we intend to employ an LLM as a natural language parser only, leaving reasoning to the rule-based solution. We demonstrate reliability on structured input, but observe surprisingly disappointing results on LLM-inferred structures, with performance below that of fully-neural (Xiong et al., 2024). The discrepancy leads us to further experiments and examination of error attribution in both pipelines, since the surprising dominance of LLMs over a verifiable reasoner resembles the known pattern of short-cutting. We find that without the modifications of CoT bootstrapping and graph augmentation, TG-LLM performs worse than our reasoner. Further, TG-LLM indeed shortcuts to the correct answer in 12% of the examined cases. The second point in particular is worth highlighting, and it resonates with a range of recent works that challenge the misleading nomenclature of 'reasoning' and 'thinking' that surrounds LLMs (Bao et al., 2024; Pfister & Jud, 2025; Stechly et al., 2025; Lee et al., 2024; Valmeekam et al., 2022; Shojaee et al., 2025). We highlight here specifically Bao et al. (2024) and Stechly et al. (2025), who both investigate the relationship of chain-of-thought tokens to the final answer. Although we are rather concerned with prompt content than steps of 'reasoning', we feel the overarching take-away is similar: final answer depends only *statistically* on the input and intermediate information, unlike in verifiable reasoning. We argue that the recent successes of these models in purported 'reasoning' stem from the deceptive result-oriented mindset, where correct answers are equated with evidence of cognition. As the structural limitations of LLMs are becoming increasingly widely acknowledged, we maintain that rule-based and symbolic computation is a valuable complement that can help redress these shortcomings. We want to stress that we do not see LLMs as inferior tools: neural and symbolic methods are complementary. Where LLMs may lack in logic, they supply the associative capabilities. This complementarity can be seen as somewhat analogous to the cognitive science categorisation into System 1 (slow, exact, deliberative) and System 2 (fast, approximate, associative) (Kahneman, 2011). The benefit of having access to flexible approximate reasoning is particularly salient in our chosen case study example, where the closest semantically match was indeed the correct one. While this is undesirable behavior when similar cases are distinct (birthplace can be a cue for disambiguating between individuals of the same name), it is desirable when e.g. working with potentially unreliable information. How to appropriately balance the flexibility of neural architectures with the reliable but rigid set of symbolic methods remains a challenge.

### 6.1 Limitations and future work

The first limitation to note is our use of only a single LLM architecture. Llama-2 13-B (Touvron et al., 2023), while competitive at the time of publication of the original work (Xiong et al., 2024), may be particularly prone to the errors we see, particularly short-cutting. Nevertheless, we argue that for the 'cautionary tale' spirit of this work, this single LLM is sufficient, if not in fact ideal – exactly due to its' limitations. We hypothesize that a newer, stronger model may indeed not exhibit such obvious short-cutting. That might seem reassuring, but all it would show is the lack of *detectable* guessing. The answers are still though subject to the opaque blend of data and chance, it is just that with scale, more data and more benchmarks released, LLMs are increasingly harder to catch in the act. We wonder whether this pursuit is the optimal direction for AI development. First, there are broader societal concerns of deceptively convincing approximations of thinking. Second, there is also the practical question of benefits of pushing LLMs towards imitating symbolic reasoning, rather than e.g. learning to delegate appropriately. What is the appropriate level of commonsense reasoning for LLMs to have is not a straightforward question, we merely pose it[6].

Three further limitations of the present work, as well as availability of convenient benchmark of Fatemi et al. (2024) lead us to scope a natural extension of the study. A first limitation is that our attempted proof-of-concept AIR-LLM offers no performance improvement over the purely neural solution. Secondly, manual implementation of question parsing and other utilities for specific benchmarks is time-costly. And finally, as a result of manual, benchmark-focused development, our AIR reasoner is not a universal one, so it will not be immediately transferrable to other problems / datasets.

The first of these limitations is straightforward to ameliorate; we simply need high-quality temporal graphs. Large Language Models are far better as parsers than reasoners; in fact, as already demonstrated by Xiong et al. (2024) on the TimeQA dataset, extracting reliable temporal graphs is possible, albeit with a little more effort and control than fine-tuning on a synthetic dataset. To address the second and third limitations, we intend to examine the possibility of leveraging PAL, in concert with synthetic data exactly like that of Fatemi et al. (2024). Specifically, LLMs would be tasked with writing executable Python programs to parse and answer questions using AIA logic and the Interval Python class, in a similar vein to Pan et al. (2023). Preliminary tests carried out suggest it to be a promising avenue. We expect that when data admits controllable graph complexity and length, a Neuro-symbolic approach will provide clear benefits over LLMs.

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

Table 3: Illustrative examples from each dataset. For TGQA only showing a single example question, for space considerations. Notation: 'TR-l2' and 'TR-l3' are TempReason l2 and l3 splits; 'TGQA' is the synthetic dataset of Xiong et al. (2024); 'TQA-e' and 'TQA-h' are TimeQA easy and hard splits.

| Dataset | Question | Temporal graph |
|---------|----------|----------------|
| TQA-e | 'Who was Arnulf Øverland's spouse from Jun 1945 to 1968?' | 1918 - 1939 : Arnulf Øverland's spouse is ( Hildur Arntzen ) 
 Feb 1940 - 1945 : Arnulf Øverland's spouse is ( Bartholine Eufemia Leganger ) 
 Jun 1945 - 1968 : Arnulf Øverland's spouse is ( Margrete Aamot Øverland ) |
| TQA-h | 'Ty Detmer played for which team in Nov 1992?' | 1987 - 1991 : Ty Detmer's team is ( Unknown ) 
 1992 - 1995 : Ty Detmer's team is ( Green Bay Packers ) 
 1996 - 1997 : Ty Detmer's team is ( Philadelphia Eagles ) 
 1999 - 2000 : Ty Detmer's team is ( Cleveland Browns ) 
 2001 - 2003 : Ty Detmer's team is ( Detroit Lions ) |
| TR-l2 | 'Which employer did J. Pelikan work for in Jan, 1948?' | J. Pelikan works for Valparaiso University from Jan, 1946 to Jan, 1949. 
 J. Pelikan works for Concordia Seminary from Jan, 1949 to Jan, 1953. 
 J. Pelikan works for University of Chicago from Jan, 1953 to Jan, 1962. 
 J. Pelikan works for Yale University from Jan, 1962 to Jan, 1962. |
| TR-l3 | 'Which employer did J. Pelikan work for after Concordia Seminary?' | J. Pelikan works for Valparaiso University from Jan, 1946 to Jan, 1949. 
 J. Pelikan works for Concordia Seminary from Jan, 1949 to Jan, 1953. 
 J. Pelikan works for University of Chicago from Jan, 1953 to Jan, 1962. 
 J. Pelikan works for Yale University from Jan, 1962 to Jan, 1962. |
| TGQA | 'What happened right before the event (Oliver Simmons died in Riverdale, Indiana) starts?' | (Oliver Thompson was born in Austin, Texas) starts at 1860. 
 (Oliver Simmons was born in Carlton, California) starts at 1863 
 (Oliver Thompson was married to Oliver Simmons) starts at 1897 
 (Oliver Simmons was married to Oliver Thompson) starts at 1897 
 (Oliver Thompson died in Riverdale, Indiana) starts at 1931 
 (Oliver Thompson was married to Oliver Simmons) ends at 1931 
 (Oliver Simmons was married to Oliver Thompson) ends at 1931 
 (Oliver Simmons died in Riverdale, Indiana) starts at 1948 |

# A  Appendix

## A.1  Datasets

We include a single example of a structured timeline – question pair from each dataset in Table 3. To elaborate on Section 3, we also briefly discuss these in a bit more depth.

First, TimeQA contains timelines focused on a single individual and the evolution of their relationship to other entities, e.g. marriage, employment, team membership. The two splits differ in that for the 'easy' questions (first row of Tab. 3), all that is required to answer is returning the correct node of a TG, as the date asked about is included *verbatim* in both the query and the timeline. In contrast, for the 'hard' split (second row of Tab. 3) the date that appears in the question does not appear in the graph, and appropriate temporal relations are needed to identify the answer. In this instance, the fact that node 'Green Bay Packers' *contains* Nov 1992 allows for selecting it as the correct answer. As can be seen, and perhaps reflecting the semi-automated generation of the benchmark, ground truth timelines may contain an 'Unknown' node. While in itself not an issue, the name 'Unknown' is also used as an answer when the date in the query falls outside of the ground truth TG. This makes it somewhat problematic for evaluation, particularly for inferred temporal graphs, where ambiguity is likely to happen at higher rates.

Second, TempReason contains timelines v. similar to those of TimeQA, both benchmarks being sourced from WikiData (Vrandečić & Krötzsch, 2014): a known person's temporally evolving relationship such as workplace. It uses a slightly different format for the TG (see row 3 of Tab. 3), and asks different questions, however. Recall that in Section 3 the 'l2' split was introduced as querying a time-time relationship. That is, the question contains a date, and the answer requires comparing it with the times of all nodes in a TG. This is akin to TimeQA; though the authors of Tan et al. (2023) do not distinguish implicit from explicit queries, we suspect majority are implicit, a.k.a. requiring carrying out temporal logic. In contrast, the 'l3' split (see row 4 of Tab. 3) queries a time-event relationship and thus introduces an additional dependence on the quality of the timeline: the date required for posing the question is not provided, only entity name.

Finally, TGQA contains fictional timelines created by anonymising sub-graphs from Yago11k. We choose an example that illustrates the issues of symmetric nodes (last row of Tab. 3). As can be seen, there are two separate entries for the marriage between O. Thompson and O. Simmons, distinguished by which of the persons is the subject vs. the object in the sentence. The date immediately preceding the one in question is 1931, and there are 3 'events' associated with it in the TG: 'O. Thomson died ...' starts in that year; and 'O. Thomson was married ...' and 'O. Simmons was married ...' both end. The ground truth correct answer to the question is, however, singular: it is the answer with O. Simmons as a subject.

An additional issue is also the treatment of the start and end point of an interval as separate, and more specifically the ambiguous employment of this approach. Namely, for events naturally associated with a duration, like marriage or employment, both start and end are provided, separately. On the other hand, birth and death are treated as separate nodes entirely, not as start and end points of the same event. This seems slightly problematic conceptually, and a little counter to teaching LLMs temporal logic. This also introduces an over-representation of start points vs. end points in the training data, which we hypothesise may bias the LLM in training.

## A.2 Methods

For completeness we include the full list of AIA relations. We note that although for question such as before/after it may suffice to use two of the AIA relations, for questions such as 'Which event was happening at *Time1*?' it may need as many as 5 (overlaps, overlapped by, during, starts, finishes).

Table 4: All 13 relations of Allen's Interval Algebra, composed of identity and 6 asymmetric relations with their inverse relation.

| Relation | Notation | Inverse | Illustration | Implementation |
|---|---|---|---|---|
| X before Y | $<$ | $>$ | XXX
YYY | X.end $<$ Y.start |
| X equal Y | $=$ | $=$ | XXX
YYY | X.start $==$ Y.start & X.end $==$ Y.end |
| X meets Y | $m$ | $mi$ | XXX
YYY | X.end $==$ Y.start |
| X overlaps Y | $o$ | $oi$ | XXX
YYY | X.start $<$ Y.start & X.end $>$ Y.start |
| X during Y | $d$ | $di$ | XXX
YYYYY | X.start $>$ Y.start & X.end $<$ Y.end |
| X starts Y | $s$ | $si$ | XXX
YYYYY | X.start $==$ Y.start & X.end $<$ Y.end |
| X finishes Y | $f$ | $fi$ | XXX
YYYYY | X.start $>$ Y.start & X.end $==$ Y.end |

