# OpenReview forum: "Unreasonable effectiveness of LLM reasoning: a doubly cautionary tale of temporal question-answering"
_TMLR — Accepted by TMLR_

### Review · Reviewer_fPWR · 2025-09-08

**Summary Of Contributions:**

The authors study a benchmark for temporal reasoning (building off an eval used for the prior work of TG-LLM). In TG-LLM, the temporal reasoning is split into 2 stages. One stage is a finetuned LLM that extracts a temporal graph of events, and the 2nd is a finetuned LLM that takes the temporal graph of events to answer a question.

In particular, the authors are interested in neurosymbolic approaches, so for the 2nd stage, they compare the finetuned LLM to a rule-based Python system, based on Allen's Interval Algebra. In cases of ambiguity, the rule based system returns all valid answers to the question (based on the extracted temporal graph), and it is considered a success if the intended outcome is within the output set.

The rule based system achieves a 99%+ success rate when given ground truth temporal graphs, but performs at only 20%-75% on datasets when switched to the stage 1 temporal graph extraction, indicating some temporal graphs are extracted incorrectly. Oddly, the number is sometimes worse than the original TG-LLM reported numbers. On an analysis of the results, they find that this is because the fully neural TG-LLM solution will sometimes perform a "shortcut" or educated guess on the answer to the question, that is not consistent with reasoning against the faulty extracted temporal graph. The authors argue that this is a sign that the LLM is not reasoning properly, since a truly logical reasoner should perform no better than the rules based solution.

**Audience:**

Yes

**Audience Explanation:**

Should be of interest to those more skeptical of LLMs / more interested in combining neural and symbolic approaches.

**Claims And Evidence:**

Yes

**Claims Explanation:**

I believe the presentation could be tightened up (Figure 3 is tricky to parse at first glance with all the dataset and method names abbreviated in one big table), but the central story of the paper makes sense.

I personally think there is room for a philosophical discussion on whether we want LLMs to be perfect reasoners that can be shifted off the intended answer by single issues in their initial premise (arguably this is good behavior and we shouldn't be asking LLMs to follow rule-based logic in settings where we have sufficiently good rule-based solvers), but that topic of debate is separate from whether the paper presents evidence that LLMs sometimes perform shortcuts rather than directly following reasoning. This evidence is certainly there.

**Requested Changes:**

Consider alternate ways of presenting Figure 3 (and to a lesser extent, Figure 4).

---

> ### Author Response · Authors · 2025-10-22
>
> Many thanks for the review and suggestions. To briefly address the comments:
>
> In terms of tightening up / making clearer the presentation, we have revised the paper: to address 3rd reviewer’s concerns we have expanded the results, with an eye to clarity of exposition of our main argument. We hope this resolves the issue.
>
> As for a potential discussion, this is a welcome suggestion, and an excellent (and nuanced) question. We (perhaps obviously) lean towards the side of *not* wanting LLMs to attempt to imitate symbolic / rule-based systems; but should that mean entirely forgoing logical capabilities in LLMs? Although a proper philosophical discussion may require a different format, we modify our discussion to pose it.

---

### Review · Reviewer_uxRb · 2025-09-09

**Summary Of Contributions:**

The paper is a case study about temporal reasoning and LLMs. The authors build a symbolic system called Allen's Interval Reasoner (AIR) based on Allen's Interval Algebra, then they extend the study of Xiong et al. (2024). This study first extracts temporal graphs from the input statements, then reasons over these using LLMs to answer questions like "Who was John's wife before Alice?". The authors substitute the LLM in the second part with the symbolic AIR.

As the symbolic AIR achieves near-perfect results on the ground truth graphs, the expectation is that it will also improve performance on the inferred (extracted) graphs. However, this turns out not to be the case: the Neuro-Symbolic approach underperforms. The authors hypothesize that this is due to the LLMs using short-cutting instead of proper reasoning, and thus providing good answers even for inferred temporal graphs which contain errors.

To confirm this, the authors run a case study on the TGQA dataset where they rerun the second phase of Xiong et al. without CoT bootstrapping or temporal graph augmentation. They also run an experiment where they provide the ground truth graphs, and show that AIR performs almost perfectly, while the LLM does not. They also show that for 10% of the dataset, the LLM answers correctly on incorrect temporal graphs.

## Strengths

The paper is an instructive and interesting tale about the unexpected outcome while replacing LLMs with a perfect symbolic module. It is easy to read and its tone is entertaining.

## Weaknesses

I have some concerns about AIR and the evaluations, and I also think the paper should be more self-contained. I detail these below.

**Additional Comments:**

- Why was TGQA selected for the short-cutting case study? Wouldn't a dataset where TG-LLM is better than AIR-LLM by a larger margin be more suitable?
- I did not understand this sentence: "Importantly, the creation includes a convenient step of selecting sets of facts connected by the subject, which means the temporal graph is already provided with the data."
- LLMs_daily_arxiv was last updated more than a year ago, it is probably not useful to include its link

**Audience:**

Yes

**Audience Explanation:**

I believe that the topic of the paper is current and interesting for TMLR readers.

**Broader Impact Concerns:**

I don't think the paper has ethical concerns. Limitations are addressed in their own section.

**Claims And Evidence:**

No

**Claims Explanation:**

I have some concerns about the claims:
- I think the results could be demonstrated to hold across multiple LLMs. Newer open-weight models are available that are similar in size to Llama 2 13B.
- The function which implements the appropriate temporal relation is selected using heuristics, like whether "before" appears in the text of the query or not. This raises the question of how dependent performance is on these heuristics, and how tailored are these heuristics to the datasets.
- I'm not sure that the heuristic of always selecting the chronologically last interval in AIR is suitable for all scenarios.
- 3.4 states that an answer is deemed correct if it is included in the set of answers identified by AIR. This seems a significant relaxation, and I wonder if it's partly why AIR achieves near perfect results on the ground truth temporal graphs.
- The paper states in 4.2 that because the performance of the LLM worsens if we remove the bootstrapped CoT and data augmentation of Xiong et al. 2024, the LLM must be short-cutting. I don't see why this follows. In general, I'm not sure that the worse performance of the LLM on TG-true is necessarily due to short-cutting.

**Requested Changes:**

- I believe that the paper should be more self-contained and less dependent on Xiong et al. 2024. Important concepts like the CoT bootstrapping, temporal graph augmentation, and the exact methodology should be explained. I had to read the original paper to understand this one.
- In 3.3, it is not clear whether a single adapter was used or multiple ones. In general I think this section could be clarified and explained in more detail.
- The row labels of Figure 3 sometimes do not match the ones in the explanation and caption. TG-pred-score is not explained in the caption.

---

> ### Author Response · Authors · 2025-10-22
>
> Many thanks for the review and suggestions. We have now revised the paper, and since a number of areas are now changed, we’d like to respond to each point.
>
> In response to concerns regarding clear evidence:
> * Newer LLMs: First, we hadn’t considered it vital, given that our main aim wasn’t and isn’t assessing the magnitude or scale of the problem, but rather highlighting / calling attention to the structural issue of LLMs (or more precisely, LLMs pushed towards scoring well on reasoning tasks + falling prey to Goodhart’s law). Second, we feel that our point remains whether a newer LLM shows similarly flawed behaviour (short-cutting) - or not. To elaborate on the latter: if a newer LLM extracts graphs much better, then AIR-LMs performance will increase. Question remains to what degree, but if AIR-LM tops fully neural solution, it confirms our original hypothesis, that a NeSy solution is better than fully neural. If AIR and LLM are on par, and there is no detectable shortcutting, that arguably would make our case a bit more difficult, rhetorically speaking: why go to the trouble of NeSy if LLMs are getting good enough to do as good a job? Well, in our view precisely because from experiments on older, weaker models we know they are not reliable reasoners + lack of evidence of guessing is not evidence of reasoning. In fact, the ongoing race to beat new benchmarks makes it increasingly _seem_ like LLMs are reasoners, when they are in fact merely ‘grokkers’ (a.k.a. skilled guessers without deeper understanding). This is a pernicious situation to be in, when symbolic methods are undeniably far more effortful than training bigger and bigger networks.
> We expand the discussion with these points.
> * Question matching heuristics: These heuristics are indeed tailored to the datasets. Further, the benchmarks have formulaic question formats that are  easy to design heuristics around. This is a significant limitation of AIR & we are not proposing AIR-LM as it stands as an attractive alternative to a fully neural solution, merely an instructional verifier to expose LLMs short-cutting. In the revised version we make it clearer to the reader.
> * Suitability of selecting chronologically last: No, it won’t be suitable for all scenarios. Similarly to the question above regarding question matching heuristics, we merely develop methods for the benchmarks. Perhaps somewhat ironic, having just invoked Goodhart’s law, but we hope sufficient for a cautionary case study.
> * Relaxed Exact Match: Many thanks for this question, it prompted a revision of our methods and made us realise that relaxed EM is indeed overly lenient. We had introduced it as there are many questions in one of the benchmarks that are wrongly assigned a single true answer. We now instead remove such questions (as described in Datasets section) and we have reevaluated results with EM instead. We have also revised the pipeline for inadequacies that the relaxed EM may have inadvertently glossed over.
> As can be seen in the revised draft, results do not change materially, nor the core message.
> * Bootstrapping removal vs short-cutting: I think this may be a matter of imprecise phrasing on our part, neither is proof of short-cutting. We’ve rephrased now to avoid confusion, in addition to expanding the analyses.
>
> In response to requested changes:
> * We now go into more detail on the original study.
> * We revise this section to clarify that a separate adapter is trained for each of TempReason, TGQA and TimeQA.
> * Many thanks for spotting these, revised now.
>
> Finally, on additional comments:
> * Indeed, any of the other two datasets would have been an 'easier target'. Primarily we did this for computational reasons, as TGQA  is smaller. Secondarily, we wanted to investigate TGQA - with TempReason and TimeQA we see a large proportion of malformed extracted graphs, which is in some sense less interesting than the more subtle failures of TGQA.
> * We now elaborate on datasets in more detail in both the main and the Appendix to hopefully clarify; although WikiData is not as useful as Yago11k for temporal knowledge graphs, it is more structured than free text Wikipedia and allows selection of temporal facts about single subjects e.g. persons, organisations.
> * Indeed, we haven't noticed; we now remove that footnote.

---

> > ### Comment · Reviewer_Apaj · 2025-11-12
> >
> > Thank you very much for your detailed changes. I believe my concerns are addressed.

---

### Review · Reviewer_Apaj · 2025-09-25

**Summary Of Contributions:**

This paper studies temporal logic in LLMs. The paper studies datasets where temporal reasoning is done through building a temporal graph structure and then using the graph to answer the temporal reasoning query. Specifically, the authors investigate behavior of rule-based & LLM systems using predicted & ground truth temporal graphs. They find:
* "no improvement over the reported bootstrapped CoT answers, yet a marked improvement over ‘vanilla’ CoT"
* "short-cutting is taking place in the LLM ... in 10% of the dataset".

This analysis is important for our understanding of what is hard about temporal reasoning for LLMs. It considers four datasets and several different settings of LLM & rule-based approaches.

It informs the reader about tradeoffs and design decisions of temporal reasoning datasets.

**Audience:**

Yes

**Audience Explanation:**

Yes, it intersects LLMs / Reasoning / Temporal - focused research.

**Claims And Evidence:**

No

**Claims Explanation:**

My primary concern with this paper is the presentation of the material and results.

Perhaps, I am coming from this as someone who is not extraordinarily well versed in temporal reasoning. However, as I understand things, Figure 4 B is the heart of the paper's core "Unreasonable effectiveness of LLM reasoning" claims. To me, this buries the core analysis too much, and leaves too many questions set-up by the rest of the paper, only to be summarized in such a small way (without additional analysis).

In more detail my concerns (which, as an aside, I believe are fully addressable via revision).

1) Both the introduction & related work section, give us very brief whirlwind tours of what has been done related to temporal reasoning. This is helpful, but it is not helpful to get us set-up for the method section of the paper. I think the intro should read much more focused on addressing the unreasonable effectiveness question, and the techniques used for that analysis that will show up in the methodology section. This will focus the reader, help them understand what is important in the paper and what is helpful context but not part of the main claim. Then the current related work section can come later in the paper. The intro should end setting up the reader with exactly the questions to be explored in the empirical analysis.
2) The Methods section is really a methods and data section. I think the paper would be stronger if the Datasets were given their own section, say section 2 (and Methods section 3). In the dataset presentation, rather than relying on the reader to go look up a lot of the details of the datasets, the presentation would be improved if, for each dataset, we are provided: 1-3 examples, and a structured table, which says: how many query, what type of query, what kind of passage, what is sota-method for that dataset. Also, when using phrases like "time-sensitive facts" without an example, it is hard for the reader to fully understand what is meant given the ambiguity of the phrase. I assume it means facts for which the time (either mentioned in the question or in context) changes the correct answer.
3) As I understand, only a small subset of the AIA rules, and so while informative, the general presentation does not drive me closer to understanding the unreasonableness claim. I am all for the general presentation, but as it is now, it doesn't bring me closer to the claim, just providing additional context.
4) I think it is important to consider how much the choice of model impacts the temporal reasoning ability. Or maybe a better way to say this is -- what parts of the claim of unreasonableness are model specific and what parts are not. Understanding how to tease these apart, I think should be a larger part of the presentation.
5) I think Figure 4B result should be given more real-estate in the paper given its importance to the claim. It should also be subject to more quantitative analysis. E.g., seeing some examples, seeing how performance breaks down for different properties of queries etc.

**Requested Changes:**

Please see answer above.

---

> ### Author Response · Authors · 2025-10-22
>
> Many thanks for the review and suggestions. We hope the revised draft addresses the concerns sufficiently, but to respond on each point individually:
> 1. We now go into a little bit more detail in the Introduction on ‘catching the LLM on its bluff’. Although we aren’t aware of many studies doing similar examinations, the few that do should be elaborated on in more detail.
> 2. Datasets are now their own section, with a summary table and more in-depth description. We also provide examples in the Appendix.
> 3. We assume the suggestion would be to delegate the full table to the Appendix, which we now do. We retain an abbreviated version for illustrative purposes.
> 4. Regarding the choice of model, we would like to highlight our response to the 2nd reviewer; we now discuss the issue of model choice and what it means for our argument - and more generally.
> 5. We have now expanded the results and thank for the suggestion.

---

### Author Response · Authors · 2025-10-22
**Thanking our reviewers and a general note on the revised submission**

Kind thanks to each of our reviewers for insightful questions and constructive feedback, these have been very helpful. We have revised our work substantially. We would like to highlight specifically: expanded detail on the benchmark datasets and the original study of Xiong et al.; revision of the pipeline for exact match (rather than its’ ‘relaxed’ version) and expanded results section; and a deeper discussion on more powerful LLMs and their reasoning-imitative capabilities.

---

### Decision · Action_Editor_wbvS · 2025-11-22

**Recommendation:** Accept as is

**Additional Comments:**

I agree with reviewer Apaj that the intro is a bit unfocused and meandering. It introduces grand questions about reasoning in LLMs, but I would say it's unclear why temporal logic is a necessary or salient environment to study them in.

Furthermore, issues around shortcutting in QA datasets have been extensively studied in past work. Most of the high-level conclusions from this paper about capabilities of LLMs for reasoning (that they identify heuristics, can do symbolic reasoning somewhat but not perfectly, etc.) broadly follow those in past work. Nevertheless, the narrative presented here is interested and mostly correct, so according to the criteria of TMLR, this paper is above the bar.

**Audience:**

Yes

**Audience Explanation:**

Yes, some in TMLR's audience will find this paper interesting, due to its commentary on relevant contemporary issues of reasoning in LLMs.

**Claims And Evidence:**

Yes

**Claims Explanation:**

This paper explores temporal question-answering, a reasoning problem that admits two relevant approaches. First, questions can be answered end-to-end with neural models like LLMs. Second, a scenario can be parsed into a temporal graph and answered symbolically.

The paper claims:

1. To show that a rule-based reasoner doesn't outperform the neural solution

2. That this is due to a performance limit from information extraction

3. That LLMs are offsetting errors made in symbolic reasoning with shortcut reasoning

These claims are largely substantiated, based on the reviewers' assessment and my reading of the paper. A chief lingering issue is the generality of the results across datasets and settings, as well as to other LLMs; however, I feel this is given appropriate caveats in the paper.